# Fair algorithms for selecting citizens' assemblies

Bailey Flanigan[1✉], Paul Gölz[1✉], Anupam Gupta[1], Brett Hennig[2] & Ariel D. Procaccia[3✉]

Globally, there has been a recent surge in 'citizens' assemblies'[1], which are a form of civic participation in which a panel of randomly selected constituents contributes to questions of policy. The random process for selecting this panel should satisfy two properties. First, it must produce a panel that is representative of the population. Second, in the spirit of democratic equality, individuals would ideally be selected to serve on this panel with equal probability[2,3]. However, in practice these desiderata are in tension owing to differential participation rates across subpopulations[4,5]. Here we apply ideas from fair division to develop selection algorithms that satisfy the two desiderata simultaneously to the greatest possible extent: our selection algorithms choose representative panels while selecting individuals with probabilities as close to equal as mathematically possible, for many metrics of 'closeness to equality'. Our implementation of one such algorithm has already been used to select more than 40 citizens' assemblies around the world. As we demonstrate using data from ten citizens' assemblies, adopting our algorithm over a benchmark representing the previous state of the art leads to substantially fairer selection probabilities. By contributing a fairer, more principled and deployable algorithm, our work puts the practice of sortition on firmer foundations. Moreover, our work establishes citizens' assemblies as a domain in which insights from the field of fair division can lead to high-impact applications.

In representative democracies, political representatives are usually selected by election. However, over the past 35 years, an alternative selection method has been gaining traction among political scientists[2,6,7] and practitioners[1,8–10]: 'sortition', which is the random selection of representatives from the population. The chosen representatives form a panel—usually known as a citizens' assembly—that convenes to deliberate on a policy question. (Such panels also go by other names; our work applies to all panels in the broader category of 'deliberative minipublics'[11].) Citizens' assemblies are now being administered by more than 40 organizations in over 25 countries[12]; one of these organizations—the Sortition Foundation in the UK—recruited 29 panels in 2020. Although many citizens' assemblies are initiated by civil-society organizations, they are also increasingly being commissioned by public authorities on municipal, regional, national and supranational levels[1]. Notably, since 2019, two Belgian regional parliaments have internally established permanent sortition bodies[13,14]. The growing use of citizens' assemblies by governments is giving the decisions of these assemblies a more direct path to affecting policy. For example, two recent citizens' assemblies commissioned by the national legislature of Ireland led to the legalization of same-sex marriage and abortion[15].

Ideally, a citizens' assembly selected using sortition acts as a microcosm of society: its participants are representative of the population, and thus its deliberation simulates the entire population convening 'under conditions where it can really consider competing arguments and get its questions answered from different points of view'[16]. However, whether this goal is realized in practice depends on exactly how assembly members are chosen.

Panel selection is generally done in two stages: first, thousands of randomly chosen constituents are invited to participate, a subset of whom opt into a 'pool' of volunteers. Then, a panel of prespecified size is randomly chosen from this pool using some fixed procedure, which we term a 'selection algorithm'. As the final and most complex component of the selection process, the selection algorithm has great power in deciding who will be chosen to represent the population. In this Article, we introduce selection algorithms that preserve the key desirable property pursued by existing algorithms, while more fairly distributing the sought-after opportunity[17–20] of being a representative.

To our knowledge, all of the selection algorithms previously used in practice (Supplementary Information section 12) aim to satisfy one particular property, known as 'descriptive representation' (that the panel should reflect the composition of the population)[16]. Unfortunately, the pool from which the panel is chosen tends to be far from representative. Specifically, the pool tends to overrepresent groups with members who are on average more likely to accept an invitation to participate, such as the group 'college graduates'. To ensure descriptive representation despite the biases of the pool, selection algorithms require that the panels they output satisfy upper and lower 'quotas' on a set of specified features, which are roughly proportional to the population rate of each feature (for example, quotas might require that a 40-person panel contain between 19 and 21 women). These quotas are generally

[1]Computer Science Department, Carnegie Mellon University, Pittsburgh, PA, USA. [2]Sortition Foundation, Cambridge, UK. [3]School of Engineering and Applied Sciences, Harvard University, Cambridge, MA, USA. ✉e-mail: bflaniga@cs.cmu.edu; pgoelz@cs.cmu.edu; arielpro@seas.harvard.edu

imposed on feature categories delineated by gender, age, education level and other attributes that are relevant to the policy issue at hand. In Supplementary Information section 3, we demonstrate that quota constraints of this form are more general than those that are achievable via 'stratified sampling', which is a technique that is often used for drawing representative samples.

Selection algorithms that pre-date this work focused only on satisfying quotas, leaving unaddressed a second property that is also central to sortition: that all individuals should have an equal chance of being chosen for the panel. Several political theorists present equality of selection probabilities as a central advantage of sortition, and stress its role in promoting ideals such as equality of opportunity[2,21], democratic equality[16,21–23] and allocative justice[23,24]. Engelstad, who introduced an influential model of the benefits of sortition, argues that this form of equality constitutes '[t]he strongest normative argument in favour of sortition'[25] (for more details on desiderata from political theory, see Supplementary Information section 4). In addition to political theorists, major practitioner groups have also advocated for equal selection probabilities[4,26]. However, these practitioners face the fundamental hurdle that, in practice, the quotas almost always necessitate selecting people with somewhat unequal probabilities, as individuals from groups that are underrepresented in the pool must be chosen with disproportionately high probabilities to satisfy the quotas. Two previous papers[27,28] have suggested mathematical models in which selection algorithms can reconcile equal selection probabilities with representativeness, but both of these studies make assumptions that are incompatible with current practice (Supplementary Information section 5).

Although it is generally impossible to achieve perfectly equal probabilities, the reasons to strive for equality also motivate a more gradual version of this goal: making probabilities as equal as possible, subject to the quotas. We refer to this goal as 'maximal fairness'. We find that our benchmark (a selection algorithm representing the previous state of the art) falls far short of this goal, giving volunteers markedly unequal probabilities across several real-world instances. This algorithm even consistently selects some types of volunteer with near-zero probability, and thus excludes them in practice from the chance to serve. We further show that, in these instances, it is possible to give all volunteers a probability of well above zero while satisfying the quotas, demonstrating that the level of inequality produced by the benchmark is avoidable.

In this Article, we close the gaps we have identified, both in theory and in practice. We first introduce not only a selection algorithm that achieves maximal fairness, but also a more general algorithmic framework for producing such algorithms. Motivated by the multitude of possible ways to quantify the fairness of an allocation of selection probabilities, our framework gives a maximally fair selection algorithm for any measure of fairness with a particular functional form. Notably, such measures include the most prominent measures from the literature on fair division[29,30], and we show that these well-established metrics can be applied to our setting by casting the problem of assigning selection probabilities as one of fair resource allocation (Supplementary Information section 9). Then, to bring this innovation into practice, we implement a deployable selection algorithm that is maximally fair according to one specific measure of fairness. We evaluate this algorithm and find that it is substantially 'fairer' than the benchmark on several real-world datasets and by multiple fairness measures. Our algorithm is now in use by a growing number of sortition organizations around the world, making it one of only a few[31–34] deployed applications of fair division.

## Algorithmic framework
### Definitions
We begin by introducing necessary terminology, which we illustrate with an example in Supplementary Information section 1. We refer to the input to a selection algorithm—a pool of size $n$, a set of quotas

and the desired panel size $k$—as an 'instance' of the panel selection problem. Given an instance, a selection algorithm randomly selects a 'panel', which is a quota-compliant set of $k$ pool members. We define the 'output distribution' of the algorithm for an instance as the distribution that specifies the probabilities with which the algorithm outputs each possible panel. Then, the 'selection probability' of a pool member is the probability that they are on a panel randomly drawn from the output distribution. We refer to the mapping from pool members to their selection probabilities as the 'probability allocation', which we aim to make as fair as possible. Finally, a 'fairness measure' is a function that maps a probability allocation to a fairness 'score' (for example, the geometric mean of probabilities, of which higher values correspond to greater fairness). An algorithm is described as 'optimal' with respect to a fairness measure if, for any instance, the fairness of the probability allocation of the algorithm is at least as high as that of any other algorithm.

### Formulating the optimization task
To inform our approach, we first analysed algorithms that pre-dated our own. Those algorithms that we have seen in use all have the same high-level structure: they select individuals for the panel one-by-one, and in each step randomly choose whom to add next from among those who—according to a myopic heuristic—seem unlikely to produce a quota violation later. As finding a quota-compliant panel is an algorithmically hard problem (Supplementary Information section 6), it is already an achievement that these simple algorithms find any panel in most practical instances. However, owing to their focus on finding any panel at all, these algorithms do not tightly control which panel they output or, more precisely, their output distribution (the probabilities with which they output different panels). Because the output distribution of an algorithm directly determines its probability allocation, the probability allocations of existing algorithms are also uncontrolled, which leaves room for them to be highly unfair.

In contrast to these existing algorithms, which have output distributions that arise implicitly from a sequence of myopic steps, the algorithms in our framework (1) explicitly compute a maximally fair output distribution and then (2) sample from that distribution to select the final panel (Fig. 1). Crucially, the maximal fairness of the output distribution found in the first step makes our algorithms optimal. To see why, note that the behaviour of any selection algorithm on a given instance is described by some output distribution; thus, as our algorithm finds the fairest possible output distribution, it is always at least as fair as any other algorithm.

As step (2) of our selection algorithm is simply a random draw, we have reduced the problem of finding an optimal selection algorithm to the optimization problem in step (1)—finding a maximally fair distribution over panels. To fully specify our algorithm, it remains only to solve this optimization problem.

### Solving the optimization task
A priori, it might seem that computing a maximally fair distribution requires constructing all possible panels, because achieving optimal fairness might necessitate assigning non-zero probability to all of them. However, such an approach would be impracticable, as the number of panels in most instances is intractably large. Fortunately, because we measure fairness according to individual selection probabilities only, there must exist an 'optimal portfolio'—a set of panels over which there exists a maximally fair distribution—containing few panels (by Carathéodory's theorem, as discussed in Supplementary Information section 7). This result brings a practical algorithm within reach, and shapes the goal of our algorithm: to find an optimal portfolio while constructing as few panels as possible.

We accomplish this goal using an algorithmic technique known as 'column generation', where, in our case, the 'columns' being generated correspond to panels (a formal description is provided in Supplementary Information section 8). As shown in Fig. 1, our algorithms find an

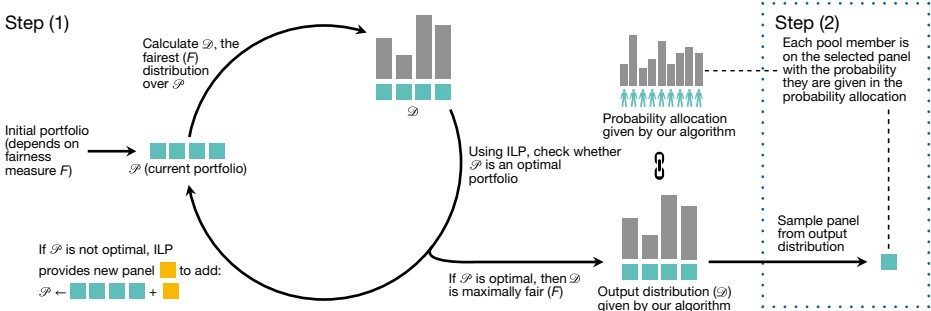

**Fig. 1 | Algorithm optimizing a fairness measure F.** Step (1): construct a maximally fair output distribution $\mathscr{D}$ over an optimal portfolio $\mathscr{P}$ of quota-compliant panels (denoted by coloured boxes), which is done by iteratively building an optimal portfolio of panels and computing the fairest distribution over that portfolio. Step (2): sample the distribution to select a final panel.

optimal portfolio by iteratively building a portfolio of panels $\mathscr{P}$. In each iteration, a panel is chosen to be added to $\mathscr{P}$ via the following two steps: (1a) finding the optimal distribution $\mathscr{D}$ over only the panels currently in $\mathscr{P}$ and (1b) adding a panel to $\mathscr{P}$ that—on the basis of the gradient of the fairness measure—will move the portfolio furthest towards optimality. This second subtask makes use of integer linear programming, which we use to generate quota-compliant panels despite the theoretical hardness of the problem. Eventually, the panel with the most promising gradient will already be in $\mathscr{P}$, in which case $\mathscr{P}$ is provably optimal, and $\mathscr{D}$ must be a maximally fair distribution. In practice, we observe that this procedure terminates after few iterations.

Our techniques extend column generation methods that are typically applied to linear programs, allowing them to be used to solve a large set of convex programs (Supplementary Information section 8.1). This extension allows our framework to be used with a wide range of fairness measures—essentially any for which the fairest distribution over a portfolio can be found via convex programming. Supported measures include those most prominent in the fair division literature: egalitarian welfare[35], Nash welfare[30], Gini inequality[36,37] and the Atkinson indices[37,38]. Our algorithmic approach also has the benefit of easily extending to organization-specific constraints beyond quotas; for example, practitioners can prevent multiple members of the same household from appearing on the same panel. Owing to its generality, our framework even applies to domains outside of sortition, such as the allocation of classrooms to charter schools[39] or kidney exchange[40] (Supplementary Information section 8.2).

## Deployable selection algorithm

To bring fair panel selection into practice, we developed an efficient implementation of a specific maximally fair selection algorithm, which we call LEXIMIN (defined in Supplementary Information section 10). LEXIMIN optimizes the well-established fairness measure leximin[30,39,41], which is sensitive to the very lowest selection probabilities. In particular, leximin is optimized by maximizing the lowest selection probability, and then breaking ties between solutions in favour of probability allocations with highest second-lowest probability, and so on. This choice of fairness measure is motivated by the fact that—as we show here and in Supplementary Information section 13—LEGACY (the algorithm used by the Sortition Foundation before their adoption of LEXIMIN) gives some pool members a near-zero probability when much more equal probabilities are possible. This type of unfairness is especially pressing because if it consistently affected pool members with particular combinations of features, these individuals and their distinct perspectives would be 'systematically excluded from participation'[42], which runs counter to a key promise of random selection.

To increase the accessibility of LEXIMIN, we have made its implementation available through an existing open-source panel selection tool[43] and on https://panelot.org/[44], a website on which anyone can run the algorithm without installation. LEXIMIN has since been deployed by several organizations, including Cascadia (USA), the Danish Board of Technology (Denmark), Nexus (Germany), of by for* (USA), Particitiz (Belgium) and the Sortition Foundation (UK). As of June 2021, the Sortition Foundation alone has already used LEXIMIN to select more than 40 panels.

## Table 1 | List of instances on which algorithms were evaluated

| Instance[a] | Pool size (n) | Panel size (k) | No. of quota categories | Mean selection probability (k/n) | LEGACY minimum probability (sampled)[b] | LEXIMIN minimum probability (exact) | Running time (LEXIMIN) |
|---|---|---|---|---|---|---|---|
| sf(a) | 312 | 35 | 6 | 11.2% | ≤0.32% | 6.7% | 20 s |
| sf(b) | 250 | 20 | 6 | 8.0% | ≤0.17% | 4.0% | 9 s |
| sf(c) | 161 | 44 | 7 | 27.3% | ≤0.15% | 8.6% | 6 s |
| sf(d) | 404 | 40 | 6 | 9.9% | ≤0.11% | 4.7% | 46 s |
| sf(e) | 1,727 | 110 | 7 | 6.4% | ≤0.03% | 2.6% | 67 min |
| cca | 825 | 75 | 4 | 9.1% | ≤0.03% | 2.4% | 7 min |
| hd | 239 | 30 | 7 | 12.6% | ≤0.09% | 5.1% | 37 s |
| mass | 70 | 24 | 5 | 34.3% | ≤14.9% | 20.0% | 1 s |
| nexus | 342 | 170 | 5 | 49.7% | ≤2.24% | 32.5% | 1 min |
| obf | 321 | 30 | 8 | 9.3% | ≤0.03% | 4.7% | 3 min |

At the request of practitioners, the topics, dates and locations of the panels are not identified.

[a]For the instances we study, panels were recruited by the following organisations. sf(a–e), Sortition Foundation; cca, Center for Climate Assemblies; hd, Healthy Democracy; mass, MASS LBP; nexus, Nexus; obf, of by for*.

[b]99% confidence, see 'Statistics' section in the Methods.

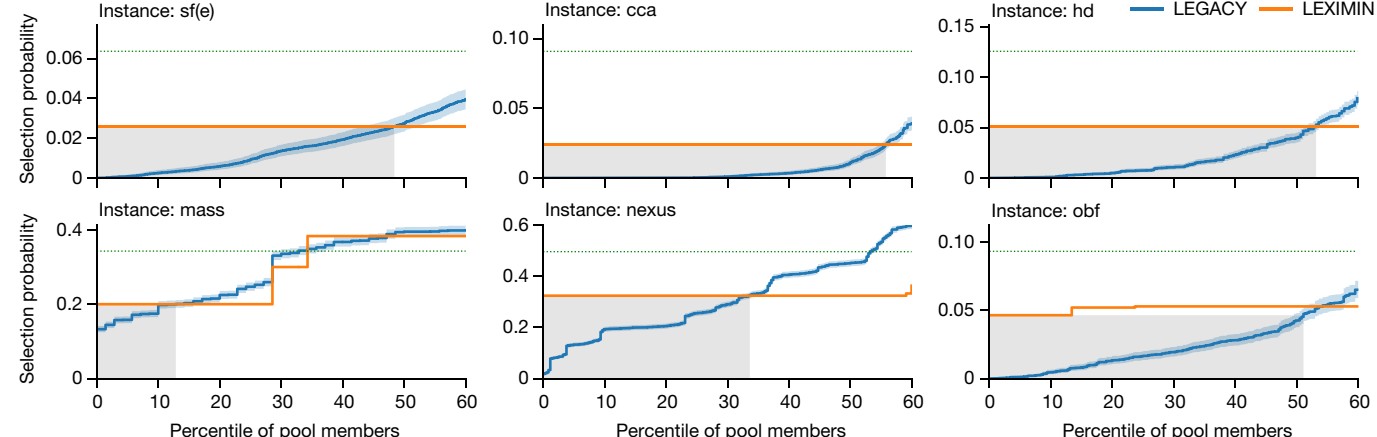

**Fig. 2 | Selection probabilities.** Selection probabilities given by LEGACY and LEXIMIN to the bottom 60% of pool members on six representative instances, in which pool members are ordered in order of increasing selection probability given by the respective algorithms. Shaded boxes denote the range of pool members with a selection probability given by LEGACY that is lower than the minimum probability given by LEXIMIN. LEGACY probabilities are estimated over 10,000 randomly sampled panels and are indicated with 99% confidence intervals (as described in 'Statistics' in the Methods). Green dotted lines show the equalized probability ($k/n$). Extended Data Figs. 1, 2 show corresponding graphs for the remaining instances and up to the 100th percentile.

We measure the effect of adopting LEXIMIN over pre-existing algorithms by comparing its fairness to that of LEGACY (described in Supplementary Information section 11). We chose LEGACY as a benchmark because it was widely used before this work, is similar to several other selection algorithms used in practice (Supplementary Information section 12) and is the only existing algorithm we found that was fully specified by an official implementation. We compare LEXIMIN and LEGACY on ten datasets from real-world panels and with respect to several fairness measures, including the minimum probability (Table 1), the Gini coefficient and the geometric mean. This analysis shows that LEXIMIN is fairer in all examined instances, and substantially so in nine out of ten.

## Effect of adopting LEXIMIN over LEGACY

We compare the fairness of LEXIMIN and LEGACY using datasets from ten citizens' assemblies, which were organized by six different sortition organizations in Europe and North America. As Table 1 shows, our instances are diverse in panel size (range of 20–170, median of 37.5) and number of quota categories (range of 4–8). On consumer hardware, the run-time of our algorithm is well within the time available in practice.

Out of concern for low selection probabilities, we first compare the minimum selection probabilities given by LEGACY and LEXIMIN, summarized in Table 1. Notably, in all instances except for 'mass' (an outlier in that its quotas only mildly restrict the fraction of panels that are feasible), LEGACY chooses some pool members with probability close to zero. We can furthermore identify combinations of features that lead to low selection probabilities by LEGACY across all instances (as described in 'Individuals rarely selected by LEGACY' in the Methods), raising the concern that LEGACY may in fact systematically exclude some groups from participation. By contrast, LEXIMIN selects no individual nearly so infrequently, with minimum selection probabilities ranging from 26% to 65% (median of 49%) of $k/n$—the 'ideal' probability individuals would receive in the absence of quotas.

One might wonder whether this increased minimum probability achieved by LEXIMIN affects only a few pool members who are most disadvantaged by LEGACY. This is not the case: as shown in Fig. 2 (shaded boxes), between 13% and 56% of pool members (median of 46%) across instances receive probability from LEGACY lower than the minimum given to anyone by LEXIMIN (Extended Data Table 2). Thus, even the first stage of LEXIMIN alone (that is, maximizing the minimum probability) provides a sizable section of the pool with more equitable access to the panel.

We have so far compared LEGACY and LEXIMIN over only the lower end of selection probabilities, as this is the range in which LEXIMIN prioritizes being fair. However, even considering the entire range of selection probabilities, we find that LEXIMIN is quantifiably fairer than LEGACY on all instances by two established metrics of fairness,

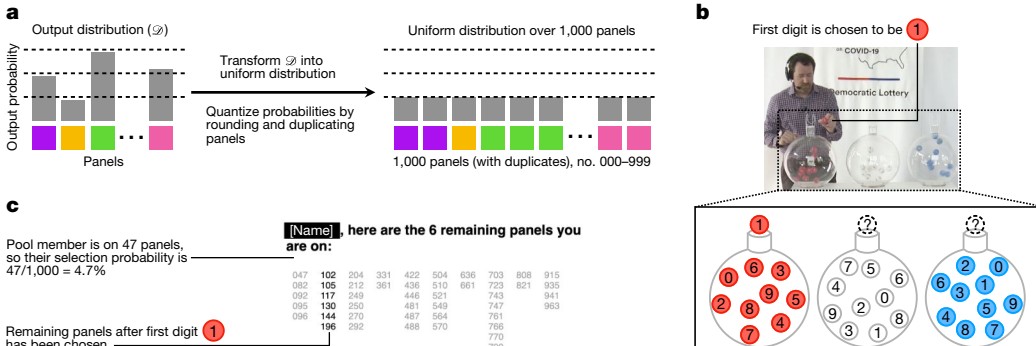

**Fig. 3 | Using LEXIMIN output to select a panel via a live uniform lottery. a**, To construct the lottery, the output distribution was transformed into a uniform distribution over 1,000 panels (numbered 000–999). **b**, During the lottery, the three digits that determined the final panel were drawn from lottery machines, making each panel observably selected with equal probability. **c**, The personalized interface (screenshot taken simultaneously with **b**) showed each pool member the number of panels out of 1,000 that they were on, allowing them to verify their own selection probabilities and those of others. Screen capture credit, of by for*.

namely the Gini coefficient and the geometric mean (Extended Data Table 1). For example, across instances (excluding the instance mass), LEXIMIN decreases the Gini coefficient—a standard measure of inequality—by between 5 and 16 percentage points (median of 12; negligible improvement for mass). Notably, the 16-point improvement in the Gini coefficient achieved by LEXIMIN on the instance 'obf' (from 59% to 43%) approximately reflects the gap between relative income inequality in Namibia (59% in 2015) and the USA (41% in 2018)[45].

## Discussion

As the recommendations made by citizens' assemblies increasingly affect public decision-making, the urgency that selection algorithms distribute this power fairly across constituents also grows. We have made substantial progress on this front: the optimality of our algorithmic framework conclusively resolves the search for fair algorithms for a broad class of fairness measures, and the deployment of LEXIMIN puts an end to some pool members being virtually never selected in practice.

Beyond these immediate benefits to fairness, the exchange of ideas we have initiated between practitioners and theorists presents continuing opportunities to improve panel selection in areas such as transparency. For example, for an assembly in Michigan, we assisted of by for* in selecting their panel using a live lottery in which participants could easily observe the probabilities with which each pool member was selected. Such lotteries represent an advance over the transparency possible with previous selection algorithms. In this instance, we found that the output distribution of LEXIMIN could be transformed into a simple lottery without a meaningful loss of fairness (Fig. 3). Further mathematical work is needed to show that this transformation can in general preserve strong fairness guarantees.

The Organisation for Economic Co-operation and Development describes citizens' assemblies as part of a broader democratic movement to 'give citizens a more direct role in [...] shaping the public decisions that affect them'[1]. By bringing mathematical structure, increased fairness and greater transparency to the practice of sortition, research in this area promises to put practical sortition on firmer foundations, and to promote the mission of citizens' assemblies to give everyday people a greater voice.

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

## Methods

### Theoretical results

The mathematical definitions and proofs supporting this Article can be found in the Supplementary Information. In Supplementary Information section 2, we formally define our model of the panel selection problem. In Supplementary Information section 6, we prove that, under widely accepted assumptions in complexity theory, panel selection algorithms cannot run in polynomial time, which justifies that our algorithms aim for acceptable running times on observed panel instances rather than for theoretical runtime guarantees. In Supplementary Information section 7, we show that Carathéodory's theorem implies the existence of small optimal portfolios, which motivates our use of column generation. Supplementary Information section 8 describes the algorithmic ideas behind our algorithmic framework and its applicability to domains outside of sortition, formally defines the framework and when it can be applied, and proves its termination and correctness. In Supplementary Information section 9, we cast the problem of panel selection into the language of fair division, which allows us to apply a range of fairness measures from the literature. We also show how each of these fairness measures can be optimized using our framework. In Supplementary Information section 10, we describe our algorithm LEXIMIN and prove its correctness. In Supplementary Information section 11, we describe the benchmark LEGACY. In Supplementary Information section 13, we construct a family of instances in which LEGACY is highly unfair even though the instances allow one to select all agents with equal probability. Finally, in Supplementary Information section 15, we analyse panel selection from an axiomatic perspective and describe why we found this approach to be less fruitful than the optimization approach we adopted in this Article.

### Individuals rarely selected by LEGACY

The empirical results in Table 1 demonstrate that, in most instances, LEGACY selects some pool members with very low probability. However, in any given citizens' assembly, this does not automatically imply that these individuals had low probability of serving on the panel. Indeed, if such an individual would have been selected by LEGACY with higher probability in most other pools that could have formed (as a result of other sets of agents being randomly invited alongside this individual), then the individual might still have had a substantial overall probability of serving on the citizens' assembly.

In this section, we show how our data suggest that this is not the case, and that some people do in fact seem to have very low likelihood overall of ending up on the panel when LEGACY is used. We make this case by demonstrating two separate points. First, we show that, across instances, LEGACY tends to give very low selection probabilities to agents who have many features that are overrepresented in the observed pool relative to the quotas. Second, we discuss why it is likely that, across possible pools for the same citizens' assembly, it is usually the same agents who have many overrepresented features. These two points, taken together, suggest that agents who have many overrepresented features in the pools we observe are rarely selected by LEGACY overall.

**Relationship between overrepresentation of features and selection probability.** To measure the relationship between the level of overrepresentation of an agent's features and that agent's selection probability by LEGACY, we first construct a simple indicator called the 'ratio product', which measures the level of overrepresentation of a given agent's set of features in the pool. The ratio product is composed of, for each of the features of an agent, the ratio between the fraction of this feature in the pool and the fraction of the quotas of the feature (specifically, the mean of lower and upper quota) in the panel. That is, if we denote the set of pool members with a feature $f$ by $N_f$ and if we

denote the lower and upper quotas of the feature by $\ell_f$ and $u_f$, respectively, then the ratio product of an agent $i$ is defined as:

$$\prod_{\text{features } f \text{ of } i} \frac{|N_f|/n}{(\ell_f + u_f)/2k}.$$

Given that the quotas are typically set in proportion to the share of the feature in the population, we say that agents with a high ratio product have many overrepresented features. Using this indicator, we find that there is a clear negative relationship in all instances between the ratio product of an individual and their selection probability by LEGACY (Extended Data Fig. 3). Most importantly, as this trend would suggest, we find that the pool members with the largest ratio products consistently have some of the lowest selection probabilities.

**The same agents probably have many overrepresented features across most possible pools.** Recall that we define an instance with respect to a single pool. However, this observed pool is only one among several hypothetical pools that could have resulted from the random process of sending out invitation letters. We define the ratio product of an agent with respect to a single instance and, therefore, a single observed pool. Then, if a different hypothetical pool (including that agent) had instead been drawn during the invitation process, the ratio product of the same agent with respect to that pool would probably be different, depending on which constituents were invited to join the pool alongside them. As the quotas and the target panel size $k$ would be the same for all these hypothetical instances, the differences in ratio product would be due to different values of $|N_f|$, for all features $f$ of the agent. Here, $|N_f|$—a random variable, the value of which is determined during the random invitation process—essentially follows a hypergeometric distribution, because it is simply the number of invitations sent to constituents who both have feature f and are willing to participate. Consequentially, all $|N_f|$ are well-concentrated, from which it follows that the ratio product of an individual should not vary much across all hypothetical pools containing them. The ratio product should be especially concentrated when all of an individual's features tend to be overrepresented, and thus all factors of the ratio product are large.

**Interpretation of results.** The analysis so far suggests that LEGACY selects individuals with many overrepresented features with low probability. Even so, one might consider the possibility that these individuals are more likely to join the pool if invited (given that they are overrepresented in the pool), and that, therefore, their lower selection probability by LEGACY in the panel-selection stage is outweighed by their higher probability of entering the pool in the pool-formation stage. This raises the question of whether the low selection probabilities given to these individuals by LEGACY are necessarily inconsistent with a scenario in which the probabilities of people going from population to panel (their 'end-to-end' probabilities[17]) are actually equal.

A back-of-the-envelope calculation suggests that this is not the case—that, in fact, the end-to-end probabilities are probably far from equal when using LEGACY. Across instances, the median ratio between the average selection probability $k/n$ and (the upper confidence bound on) the minimum selection probability given by LEGACY is larger than 100. If the selection probability of an individual conditioned on appearing in some pool is indeed 100 times lower than that of an 'average' citizen, the individual would have to enter the pool 100 times more frequently than this average citizen to serve on the panel with equal end-to-end probability. Given that average response rates are typically between 2 and 5%, someone opting into the pool 100 times more frequently than an average citizen is simply not possible.

Although we have demonstrated that LEGACY underrepresents a specific group (agents with many overrepresented features), we do not have reason to believe that LEGACY would exclude groups defined by intersections of few features (for example, 'young women' or 'conservatives

with a university degree' are the intersection of two features). In Supplementary Information section 14, we investigate the representation of such groups for one instance, 'sf(e)'. There, we find that LEGACY and LEXIMIN represent intersectional groups to similar degrees of accuracy (Extended Data Fig. 4), explore factors determining the representation of an intersectional group and describe how the accuracy of intersectional representation could be improved using our algorithmic framework.

### Instance-data preprocessing

At the request of practitioners, we pseudonymize the features of each dataset. This does not affect the analysis, as both LEGACY and LEXIMIN are agnostic to this information.

For data from Healthy Democracy (instance 'hd'), of by for* (instance 'obf') and MASS LBP (instance 'mass'), and for the instance 'sf(e)' from the Sortition Foundation, respondent data and quotas were taken without modification. For privacy reasons, pool members with non-binary gender in the instances 'sf(a)' to 'sf(d)' were randomly assigned female or male gender with equal probability. In two of these instances ('sf(a)' and 'sf(d)'), the originally used quotas were not recorded in the data, but we reconstructed them according to the procedures of the Sortition Foundation for constructing quotas from the population fractions. The panel from the Center for Climate Assemblies (instance 'cca') did not formally use upper and lower quotas; instead, exact target values for each feature were given (which could not simultaneously be satisfied) as well as a priority order over which targets were more important than others. We set quotas by identifying the minimal relaxation to the lowest-priority target that could be satisfied. For the Nexus instance (instance 'nexus'), the region of one pool member was missing and inferred from their city of residence. Because Nexus only used lower quotas, the upper quotas of each feature were set to the difference between $k$ and the sum of lower quotas of all other features of the same category. Such a change does not influence the output distribution of either LEGACY or LEXIMIN but makes the ratio product defined in 'Individuals rarely selected by LEGACY' above more meaningful. Because Nexus permitted $k$ to range between 170 and 175, we chose 170 to make their lower quotas as tight as possible.

### Statistics

The selection probabilities of LEXIMIN are not empirical estimates, but rather exact numbers generated by the algorithm, computed from its output distribution.

By contrast, the selection probabilities given to each agent by LEGACY (as used in the numbers in the text and tables) refer to the fraction of 10,000 sampled panels in which the agent appears (in which each sample is from a single run of LEGACY on the same instance).

In Fig. 2, Extended Data Figs. 1, 2, when plotting the line representing LEGACY, agents are sorted along the $x$ axis in order of this empirical estimate of their selection probability by LEGACY, and this is the selection probability given on the $y$ axis. As, for each agent, the number of panels on which they appear across runs of LEGACY is distributed as a binomial variable with 10,000 trials and unknown success probability, we indicate Jeffreys' intervals for each of these success probabilities (that is, selection probabilities) with 99% confidence[46]. These are confidence intervals on the selection probability of a specific agent, not on the selection probability of a specific percentile of the agents.

In addition to reporting two-sided 99% confidence intervals on each agents' selection probability by LEGACY, in Table 1, we report a 99% confidence upper bound on the minimum selection given to any agent by LEGACY per instance. We cannot simply set this upper bound equal to the smallest upper end of the two-sided confidence interval of any agent as computed above because out of these many confidence intervals, some are likely to lie entirely below the true selection probability of the respective agent. Instead, we compute the upper bound on the minimum probability using the confidence interval for a single agent, by running two independent sets of 10,000 samples: In the first set of samples (the one discussed two paragraphs prior), we identify a single agent who

was least frequently chosen to the panel in this set; then, we count how often this specific agent is selected across the second set of samples and calculate an upper bound based on a one-sided Jeffreys' interval as follows: if the specific agent was selected in $s$ out of the 10,000 panels, the confidence bound is the 99th percentile of the distribution beta($1/2 + s$, $1/2 + 10,000 - s$). (The bound would be 1 if $s = 10,000$, but this does not happen in any of the instances.) With 99% confidence, this is an upper bound on the selection probability of the specific agent, and thus also an upper bound with 99% confidence on the minimum selection probability.

As the magnitudes of the two-sided confidence intervals in Fig. 2 and Extended Data Figs. 1, 2 show, the empirical estimates we get of the selection probabilities of agents by LEGACY are likely to be close to their true values. Moreover, two of the three statistics we report are not very sensitive to sampling errors: For Gini inequality, additive errors in the estimate of selection probabilities translate into additive errors in the Gini coefficient; and, when we report the number of agents whose selection probability by LEGACY lies under the minimum selection probability of LEXIMIN, Fig. 2 and Extended Data Figs. 1, 2 show that the confidence intervals of most agents lie either below or above this threshold. Therefore, our analysis of LEGACY selection probabilities should not be substantially affected by the fact that we can only use empirical estimates of selection probabilities rather than the ground-truth selection probabilities themselves. The one exception is the geometric mean, for which the error in estimating small selection probabilities can severely affect the measure. In particular, in all instances in which one individual appeared in 0 out of 10,000 sampled panels, the geometric mean of empirical selection probabilities would be 0. Thus, when computing the geometric mean for LEGACY in Extended Data Table 1 and in the body of the Article, we erred on the side of being generous to LEGACY by setting the selection probabilities of these individuals to 1/10,000 instead of 0.

The running times of LEXIMIN were measured on a 2017 Macbook Pro with a 3.1-GHz dual-core Intel i5 processor. Although the running time should not depend on random decisions in the algorithm, the running time of calls to the optimization library Gurobi depends on how the operating system schedules different threads. Reported times are medians of three runs, and are rounded to the nearest second if below 60 s, or to the nearest minute otherwise.

### Reporting summary

Further information on research design is available in the Nature Research Reporting Summary linked to this paper.

## Data availability

The panel datasets analysed in this Article are not publicly available owing to the potential for identifying specific panels or participants. We cannot share the dataset nexus owing to agreements between Nexus and their upstream data sources. All other datasets are available from P.G. for research purposes only. Any publication of results based on these data are subject to the permission of the organizations supplying the data. For cca and hd data, publication does not require permission.

## Code availability

An implementation of our selection algorithm LEXIMIN as well as all code required to reproduce the empirical results of this Article are available at https://github.com/pgoelz/citizensassemblies-replication.

46. Brown, L. D., Cai, T. T. & DasGupta, A. Interval estimation for a binomial proportion. *Stat. Sci.* **16**, 101–117 (2001).

**Acknowledgements** We thank all the practitioners who have provided data and/or shared their insights with us, including G. Muller and M. Chang (on behalf of Cascadia Consulting Group), A. Gąsiorowska and M. Gerwin (on behalf of the Center for Climate Assemblies), D. Schecter (on behalf of Democracy R&D), Y. Dejaeghere (on behalf of G1000), J. Birkenhäger (on behalf of

IFOK), C. Ellis (on behalf of MASS LBP), C. von Blanckenburg (on behalf of Nexus), A. Cronkright and G. Zisiadis (on behalf of of by for*), P. Verpoort (on behalf of the Sortition Foundation) and S. Pek (on behalf of the University of Victoria); B. Cook, A. Kazachkov, F. Kilinc-Karzan and B. Shepherd for technical discussions; and Y. Dejaeghere, W. Flanigan, J. Gastil, M. Gray, T. Lee, L. Leopold, E. Vitercik and M. Wang for comments on the text. This work was supported by the Office of Naval Research under grant N00014-20-1-2488 (A.D.P.); by the National Science Foundation under grants CCF-1907820 (A.G.), CCF-1955785 (A.G.), CCF-2006953 (A.G.), CCF-2007080 (A.D.P.) and IIS-2024287 (P.G. and A.D.P.); by a Fannie and John Hertz Foundation Fellowship (B.F.), a JPMorgan Chase AI Research Fellowship (P.G.) and a National Science Foundation Graduate Research Fellowship (B.F.). The funders had no role in study design, data collection and analysis, the decision to publish, or preparation of the manuscript.

**Author contributions** All authors contributed to the problem formulation, and B.F., P.G., A.G. and A.D.P. did the technical (theoretical) work. B.F., P.G. and B.H. procured the data. B.F. and P.G. conceived the experiments, and P.G. implemented the experiments and the algorithm used by practitioners. B.F. and P.G. took the lead on writing the paper and supplementary materials, and all authors contributed to the editing process.

**Competing interests** B.H. is the founder and co-director of the Sortition Foundation.

**Additional information**
**Correspondence and requests for materials** should be addressed to B.F., P.G. or A.D.P.

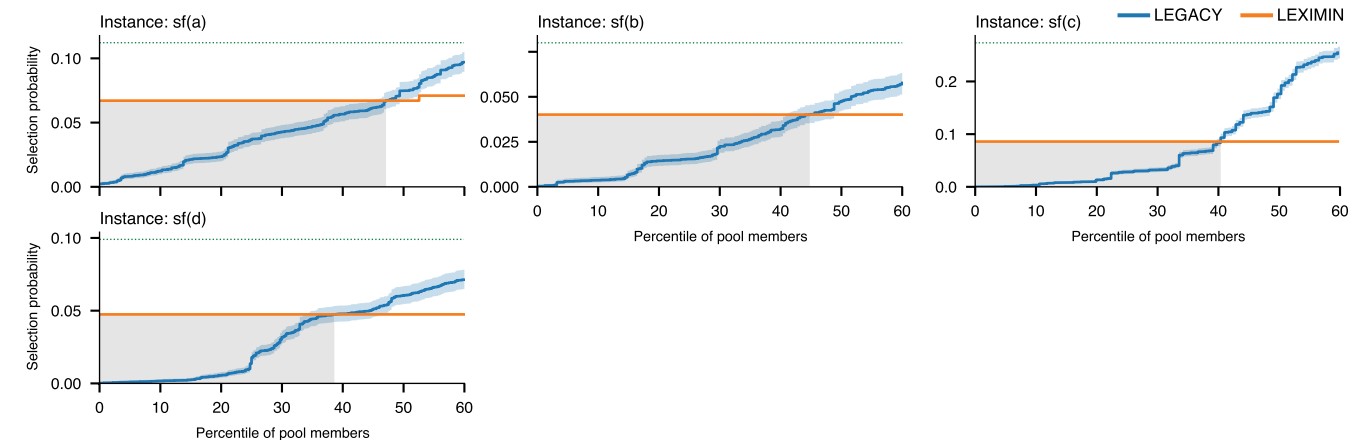

**Extended Data Fig. 1 | Selection probabilities for remaining instances.** Selection probabilities given by LEGACY and LEXIMIN to the bottom 60% of pool members on the 4 instances that are not shown in Fig. 2. Pool members are ordered across the x axis in order of increasing probability given by the respective algorithms. Shaded boxes denote the range of pool members with a selection probability given by LEGACY that is lower than the minimum probability given by LEXIMIN. LEGACY probabilities are estimated over 10,000 random panels and are indicated with 99% confidence intervals (as described in 'Statistics' in the Methods). Green dotted lines show the equalized probability ($k/n$).

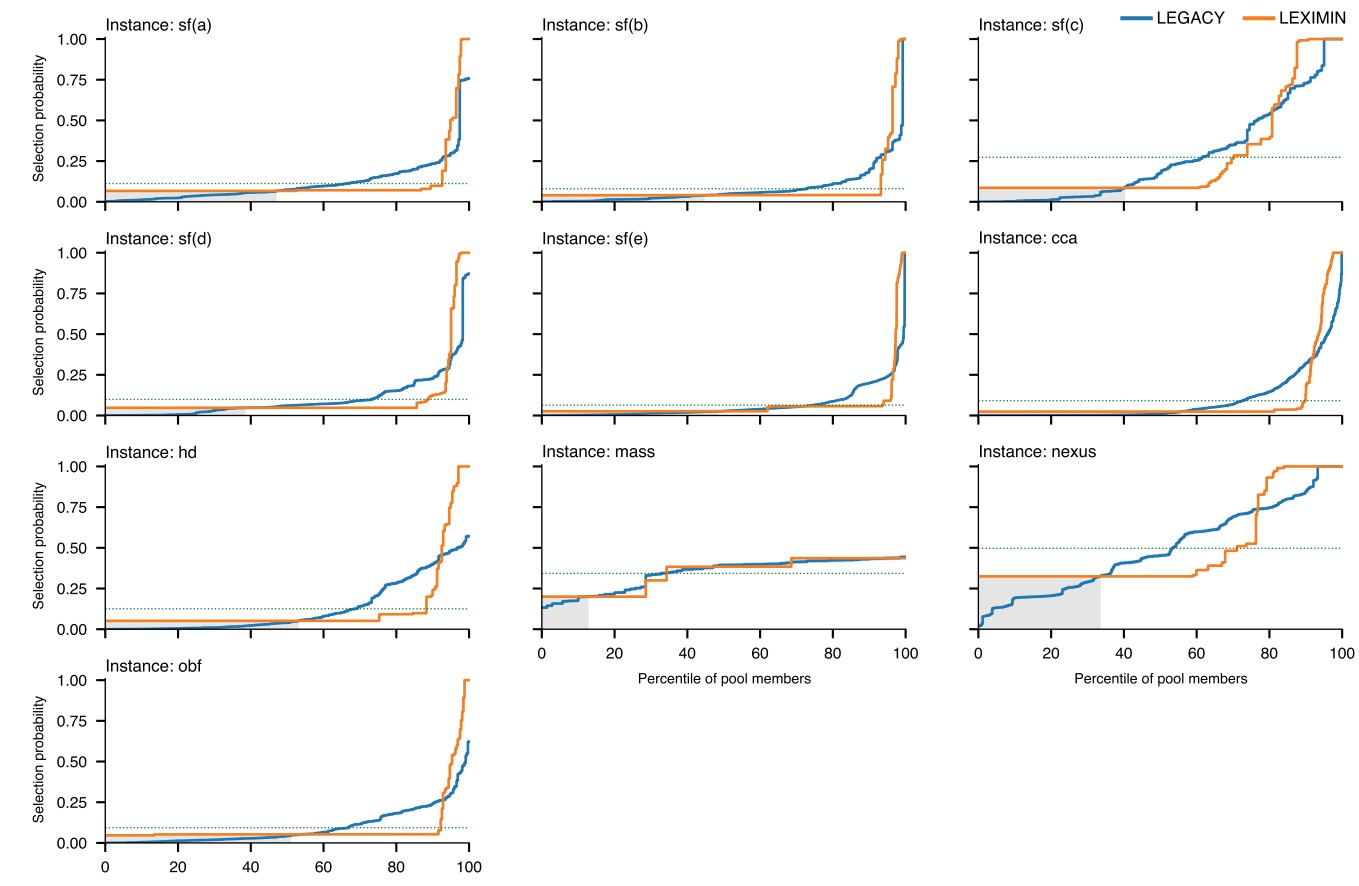

**Extended Data Fig. 2 | Selection probabilities up to the 100th percentile.** Selection probabilities given by LEGACY and LEXIMIN on all ten instances. Pool members are ordered across the *x* axis in order of increasing probability given by the respective algorithms. In contrast to Fig. 2 and Extended Data Fig. 1, this graph shows the full range of selection probabilities (up to the 100th percentile). Shaded boxes denote the range of pool members with a selection probability given by LEGACY that is lower than the minimum probability given by LEXIMIN. LEGACY probabilities are estimated over 10,000 random panels and are indicated with 99% confidence intervals (as described in 'Statistics' in the Methods). Green dotted lines show the equalized probability ($k/n$).

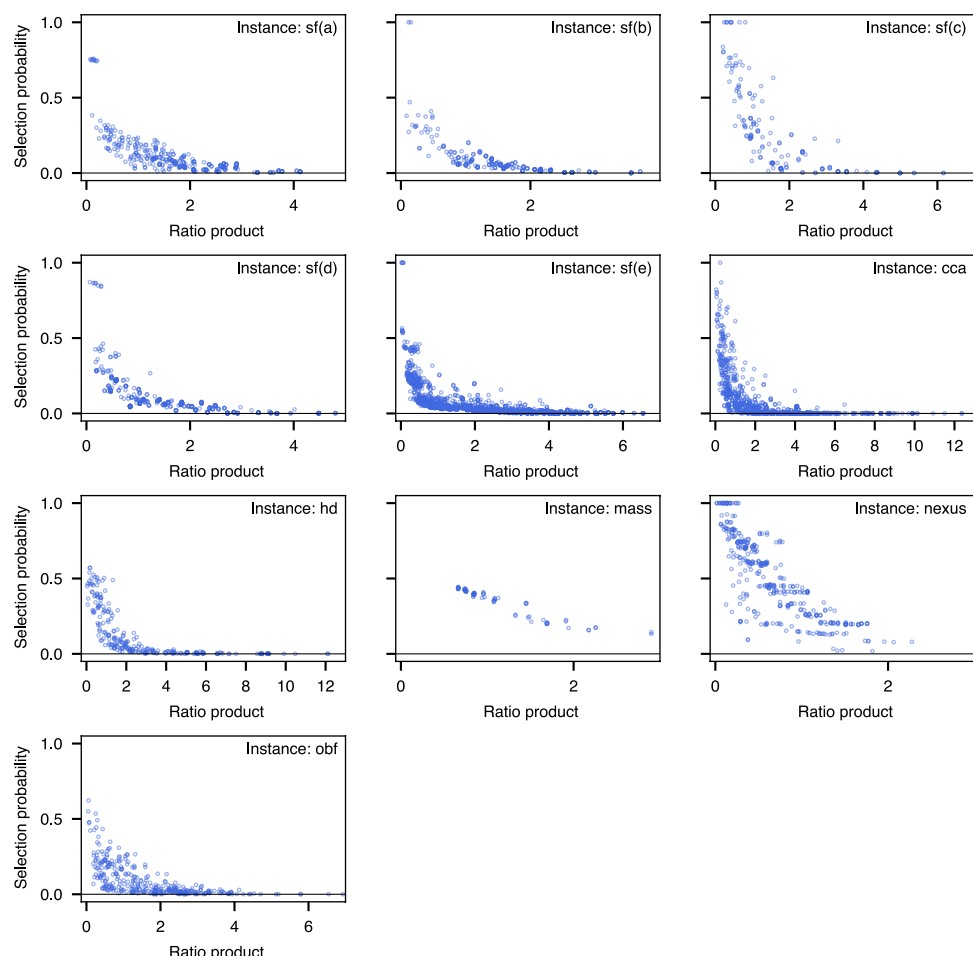

**Extended Data Fig. 3 | Overrepresentation and LEGACY selection probabilities.** Relationship between how overrepresented the features of an agent are and how likely they are to be chosen by the LEGACY algorithm. The level of overrepresentation is quantified as the ratio product (as described in 'Individuals rarely selected by LEGACY' in the Methods); agents further to the right are more overrepresented. Across instances, pool members with high ratio product are consistently selected with very low probabilities.

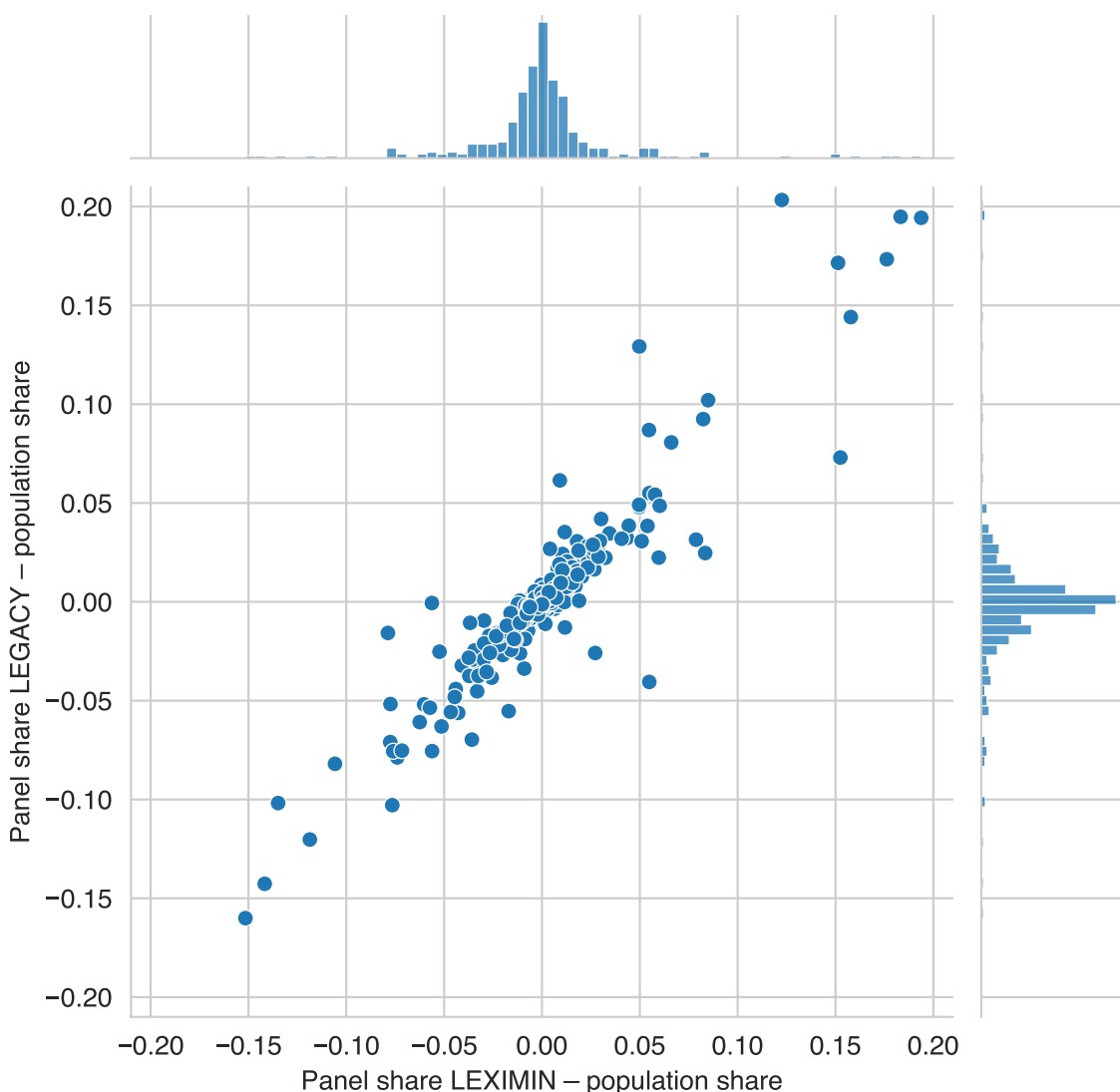

**Extended Data Fig. 4 | Representation of feature intersections.** For all intersections of two features on the instance sf(e), how far the expected number of group members selected by LEGACY or LEXIMIN differs from the proportional share in the population is shown. Although many intersectional groups are represented close to accurately, some groups are over- and underrepresented by more than 15 percentage points by either algorithm. Which groups get over- and underrepresented is highly correlated between both algorithms. Panel shares are computed for a pool of size 1,727, and population shares are based on a survey with 1,915 respondents after cleaning.

**Extended Data Table 1 | Gini coefficient and geometric mean of LEGACY and LEXIMIN**

| Instance | Gini coefficient of LEGACY (lower is fairer) | Gini coefficient of LEXIMIN (lower is fairer) | Geometric mean of LEGACY (higher is fairer) | Geometric mean of LEXIMIN (higher is fairer) |
|---|---|---|---|---|
| sf(a) | 51.2% | 37.3% | 6.5% | 8.1% |
| sf(b) | 59.6% | 47.4% | 3.5% | 4.8% |
| sf(c) | 57.0% | 52.5% | 8.3% | 16.3% |
| sf(d) | 59.3% | 48.7% | 3.5% | 6.0% |
| sf(e) | 64.4% | 51.2% | 2.2% | 3.9% |
| cca | 75.3% | 67.8% | 0.7% | 3.5% |
| hd | 64.5% | 52.9% | 3.1% | 7.3% |
| mass | 14.9% | 14.8% | 32.6% | 32.7% |
| nexus | 30.8% | 25.4% | 40.9% | 44.2% |
| obf | 58.9% | 42.7% | 3.7% | 6.2% |

Gini coefficient and geometric mean of probability allocations of both algorithms, for each instance. On every instance, LEGACY has a lower Gini coefficient and a larger geometric mean. For computing the geometric mean, we slightly correct upward empirical selection probabilities of LEGACY that are close to zero (as described in 'Statistics' in the Methods).

**Extended Data Table 2 | Share below LEXIMIN minimum probability**

| Instance | Share selected by LEGACY with probability below LEXIMIN minimum selection probability |
|----------|------------------------------------------------------------------------------------|
| sf(a) | 47.1% |
| sf(b) | 44.8% |
| sf(c) | 40.4% |
| sf(d) | 38.6% |
| sf(e) | 48.4% |
| cca | 55.8% |
| hd | 53.1% |
| mass | 12.9% |
| nexus | 33.6% |
| obf | 51.1% |

For each instance, the share of pool members selected with lower probability by LEGACY than the minimum selection probability of LEXIMIN is shown. This corresponds to the width of the shaded boxes in Fig. 2, Extended Data Figs. 1, 2.

# Reporting Summary

## Statistics

For all statistical analyses, confirm that the following items are present in the figure legend, table legend, main text, or Methods section.

| n/a | Confirmed | |
|---|---|---|
| ☐ | ☒ | The exact sample size ($n$) for each experimental group/condition, given as a discrete number and unit of measurement |
| ☐ | ☒ | A statement on whether measurements were taken from distinct samples or whether the same sample was measured repeatedly |
| ☒ | ☐ | The statistical test(s) used AND whether they are one- or two-sided<br>*Only common tests should be described solely by name; describe more complex techniques in the Methods section.* |
| ☒ | ☐ | A description of all covariates tested |
| ☒ | ☐ | A description of any assumptions or corrections, such as tests of normality and adjustment for multiple comparisons |
| ☐ | ☒ | A full description of the statistical parameters including central tendency (e.g. means) or other basic estimates (e.g. regression coefficient) AND variation (e.g. standard deviation) or associated estimates of uncertainty (e.g. confidence intervals) |
| ☒ | ☐ | For null hypothesis testing, the test statistic (e.g. $F$, $t$, $r$) with confidence intervals, effect sizes, degrees of freedom and $P$ value noted<br>*Give P values as exact values whenever suitable.* |
| ☒ | ☐ | For Bayesian analysis, information on the choice of priors and Markov chain Monte Carlo settings |
| ☒ | ☐ | For hierarchical and complex designs, identification of the appropriate level for tests and full reporting of outcomes |
| ☒ | ☐ | Estimates of effect sizes (e.g. Cohen's $d$, Pearson's $r$), indicating how they were calculated |

*Our web collection on statistics for biologists contains articles on many of the points above.*

## Software and code

Policy information about availability of computer code

| Data collection | The data was provided to us by practitioners, no software was used for data collection on our side. |
|---|---|
| Data analysis | All code is available at https://github.com/pgoelz/citizensassemblies-replication as described in the Code Availability statement and the Code and Software submission checklist. The code has not been updated since the last revision. |

For manuscripts utilizing custom algorithms or software that are central to the research but not yet described in published literature, software must be made available to editors and reviewers. We strongly encourage code deposition in a community repository (e.g. GitHub). See the Nature Portfolio guidelines for submitting code & software for further information.

## Data

Policy information about availability of data

All manuscripts must include a data availability statement. This statement should provide the following information, where applicable:
- Accession codes, unique identifiers, or web links for publicly available datasets
- A description of any restrictions on data availability
- For clinical datasets or third party data, please ensure that the statement adheres to our policy

The panel datasets analyzed in this paper are not publicly available due to privacy concerns of the sortition organizations but are available from P.G. on reasonable request. (All datasets are attached to paper submission.)

# Field-specific reporting

Please select the one below that is the best fit for your research. If you are not sure, read the appropriate sections before making your selection.

☐ Life sciences    ☐ Behavioural & social sciences    ☐ Ecological, evolutionary & environmental sciences

# Life sciences study design

All studies must disclose on these points even when the disclosure is negative.

| | |
|---|---|
| Sample size | *Describe how sample size was determined, detailing any statistical methods used to predetermine sample size OR if no sample-size calculation was performed, describe how sample sizes were chosen and provide a rationale for why these sample sizes are sufficient.* |
| Data exclusions | *Describe any data exclusions. If no data were excluded from the analyses, state so OR if data were excluded, describe the exclusions and the rationale behind them, indicating whether exclusion criteria were pre-established.* |
| Replication | *Describe the measures taken to verify the reproducibility of the experimental findings. If all attempts at replication were successful, confirm this OR if there are any findings that were not replicated or cannot be reproduced, note this and describe why.* |
| Randomization | *Describe how samples/organisms/participants were allocated into experimental groups. If allocation was not random, describe how covariates were controlled OR if this is not relevant to your study, explain why.* |
| Blinding | *Describe whether the investigators were blinded to group allocation during data collection and/or analysis. If blinding was not possible, describe why OR explain why blinding was not relevant to your study.* |

# Behavioural & social sciences study design

All studies must disclose on these points even when the disclosure is negative.

| | |
|---|---|
| Study description | *Briefly describe the study type including whether data are quantitative, qualitative, or mixed-methods (e.g. qualitative cross-sectional, quantitative experimental, mixed-methods case study).* |
| Research sample | *State the research sample (e.g. Harvard university undergraduates, villagers in rural India) and provide relevant demographic information (e.g. age, sex) and indicate whether the sample is representative. Provide a rationale for the study sample chosen. For studies involving existing datasets, please describe the dataset and source.* |
| Sampling strategy | *Describe the sampling procedure (e.g. random, snowball, stratified, convenience). Describe the statistical methods that were used to predetermine sample size OR if no sample-size calculation was performed, describe how sample sizes were chosen and provide a rationale for why these sample sizes are sufficient. For qualitative data, please indicate whether data saturation was considered, and what criteria were used to decide that no further sampling was needed.* |
| Data collection | *Provide details about the data collection procedure, including the instruments or devices used to record the data (e.g. pen and paper, computer, eye tracker, video or audio equipment) whether anyone was present besides the participant(s) and the researcher, and whether the researcher was blind to experimental condition and/or the study hypothesis during data collection.* |
| Timing | *Indicate the start and stop dates of data collection. If there is a gap between collection periods, state the dates for each sample cohort.* |
| Data exclusions | *If no data were excluded from the analyses, state so OR if data were excluded, provide the exact number of exclusions and the rationale behind them, indicating whether exclusion criteria were pre-established.* |
| Non-participation | *State how many participants dropped out/declined participation and the reason(s) given OR provide response rate OR state that no participants dropped out/declined participation.* |
| Randomization | *If participants were not allocated into experimental groups, state so OR describe how participants were allocated to groups, and if allocation was not random, describe how covariates were controlled.* |

# Ecological, evolutionary & environmental sciences study design

All studies must disclose on these points even when the disclosure is negative.

| | |
|---|---|
| Study description | *Briefly describe the study. For quantitative data include treatment factors and interactions, design structure (e.g. factorial, nested, hierarchical), nature and number of experimental units and replicates.* |
| Research sample | *Describe the research sample (e.g. a group of tagged Passer domesticus, all Stenocereus thurberi within Organ Pipe Cactus National* |

| | |
|---|---|
| Research sample | *Monument), and provide a rationale for the sample choice. When relevant, describe the organism taxa, source, sex, age range and any manipulations. State what population the sample is meant to represent when applicable. For studies involving existing datasets, describe the data and its source.* |
| Sampling strategy | *Note the sampling procedure. Describe the statistical methods that were used to predetermine sample size OR if no sample-size calculation was performed, describe how sample sizes were chosen and provide a rationale for why these sample sizes are sufficient.* |
| Data collection | *Describe the data collection procedure, including who recorded the data and how.* |
| Timing and spatial scale | *Indicate the start and stop dates of data collection, noting the frequency and periodicity of sampling and providing a rationale for these choices. If there is a gap between collection periods, state the dates for each sample cohort. Specify the spatial scale from which the data are taken* |
| Data exclusions | *If no data were excluded from the analyses, state so OR if data were excluded, describe the exclusions and the rationale behind them, indicating whether exclusion criteria were pre-established.* |
| Reproducibility | *Describe the measures taken to verify the reproducibility of experimental findings. For each experiment, note whether any attempts to repeat the experiment failed OR state that all attempts to repeat the experiment were successful.* |
| Randomization | *Describe how samples/organisms/participants were allocated into groups. If allocation was not random, describe how covariates were controlled. If this is not relevant to your study, explain why.* |
| Blinding | *Describe the extent of blinding used during data acquisition and analysis. If blinding was not possible, describe why OR explain why blinding was not relevant to your study.* |

Did the study involve field work? ☐ Yes ☐ No

## Field work, collection and transport

| | |
|---|---|
| Field conditions | *Describe the study conditions for field work, providing relevant parameters (e.g. temperature, rainfall).* |
| Location | *State the location of the sampling or experiment, providing relevant parameters (e.g. latitude and longitude, elevation, water depth).* |
| Access & import/export | *Describe the efforts you have made to access habitats and to collect and import/export your samples in a responsible manner and in compliance with local, national and international laws, noting any permits that were obtained (give the name of the issuing authority, the date of issue, and any identifying information).* |
| Disturbance | *Describe any disturbance caused by the study and how it was minimized.* |

# Reporting for specific materials, systems and methods

We require information from authors about some types of materials, experimental systems and methods used in many studies. Here, indicate whether each material, system or method listed is relevant to your study. If you are not sure if a list item applies to your research, read the appropriate section before selecting a response.

### Materials & experimental systems

| n/a | Involved in the study |
|---|---|
| ☒ | ☐ Antibodies |
| ☒ | ☐ Eukaryotic cell lines |
| ☒ | ☐ Palaeontology and archaeology |
| ☒ | ☐ Animals and other organisms |
| ☒ | ☐ Human research participants |
| ☒ | ☐ Clinical data |
| ☒ | ☐ Dual use research of concern |

### Methods

| n/a | Involved in the study |
|---|---|
| ☒ | ☐ ChIP-seq |
| ☒ | ☐ Flow cytometry |
| ☒ | ☐ MRI-based neuroimaging |

