## [Peer Review File · Nature]

Manuscript Title: Fair Algorithms for Selecting Citizens' Assemblies

Editorial Notes: *none*

Reviewer Comments & Author Rebuttals

Reviewer Reports on the Initial Version:

Referee expertise:

Referee #1: computer science

Referee #2: political science (signed review)

Referee #3: informatics

Referees' comments:

Referee #1 (Remarks to the Author):

The paper describes a new procedure for the task of sortition: selecting a sample of the population of a given size (a panel) among the willing potential participants so as to satisfy upper and lower quotas on various attributes (age, gender, education, etc.) Sortition is currently used by many organizations, but the existing algorithms are ad hoc. This submission proposes a principled way of designing sortition procedures: first, one formulates a fairness criterion wrt agents' probabilities of being selected (a natural one is to maximize the minimum probability, but others are possible) and then the algorithm outputs a sampling procedure (a distribution over a small number of panels) that optimizes this criterion.

I feel that the paper is a good fit for the journal: it addresses a problem of significant practical importance in a principled way, showing considerable improvement over existing algorithms. It makes use of real-life sortition datasets; while I am not an expert on statistics, the statistical analysis appears to be well-executed.

The paper is well-written, and the contribution is well-positioned with respect to the prior work, though I think it would be best to add details on the prior work on sortition by a subset of the authors.

What is not entirely clear to me is whether fairness towards individual agents should be the main goal: are agents eager to be selected or volunteer out of a sense of duty? Is there evidence that maximizing some form of fairness, in the spirit formalized in the paper, is viewed as desirable in the broader sortition community?

Another issue that I'd like to see discussed is intersectionality: is it possible to compare the algorithmic framework proposed in the paper to legacy algorithms in this regard? E.g., in one example considered in the paper the legacy algorithm is less likely than the fair algorithm to choose a conservative female, which is a rare combination of attributes. One would think that

including many different combinations of attributes may be a relevant goal. Can the authors comment on the behavior of their algorithms with respect to this goal?

Referee #2 (Remarks to the Author):

I'll begin this review by noting the limits of my expertise: I'm a political theorist, and so I'm not able to understand or critique the mathematics in this manuscript, beyond appreciating the natural language explanations.

Deliberative minipublics, the topic of this manuscript, are bodies of 20-500 citizens that are both representative and deliberative, usually convened to issue recommendations on a policy question. Representativeness is achieved through stratified random sampling, while deliberativeness is achieved through facilitated learning and discussion, usually over periods ranging from a weekend to as long as a year.

The importance of deliberative minipublics is that they accomplish two kinds of things that other institutions do not: they enable citizens to learn and deliberate about an issue, and they better represent publics than either elected or self-selected processes. One of the important political effects is that they bridge between popular (democratic) control and expertise. For these reasons, deliberative minipublics, while still fairly rare, are becoming much more common, mostly but not exclusively within the developed democracies. These kinds of processes will likely become more common as citizens pressure for more control, and elites seek ways of injecting expertise, deliberativeness, and the perspectives of citizens into policy processes. The importance for Nature is that deliberative minipublics increasingly look like good methods for injecting scientific expertise into policy processes, particularly in an era in which expertise viewed with skepticism by a vocal anti-elite populists (Trump, Brexiteers, etc.). All the sciences require some level of public support, and so we need to be thinking about new ways of generating that support.

So I am very pleased that Nature is considering this kind of interdisciplinary contribution to the literature and practice. The design and justification of these processes was originally driven by political theorists and democracy advocates, but their execution has required expertise in designing selection processes that constitute a body that is representative of an affected public, usually through random or stratified random selection. This paper presents an algorithm that accomplishes the selection process as fairly as possible, meaning that each person in a demographic category has a roughly equal chance of being selected. The normative justification for the algorithm is that the fairness of the selection process should underwrite its (democratic) legitimacy.

The arguments for the algorithm and its contributions to a fair selection process are nicely and clearly made. The manuscript is a very nice contribution to the literature, and will help to raise the visibility of deliberative minipublics, especially within the scientific community. The algorithm itself (assuming it withstands scrutiny from reviewers with relevant expertise) will be important to supporting the organization and practice of deliberative minipublics.

I do have some suggestions for revising the political theory that is embedded in the paper.

First, the authors don't quite have the covering terminology right. As the terms have developed, "citizens' assemblies" are one kind of "deliberative minipublic," which is the covering term. The OECD report to which the authors refer at the beginning of the paper calls these processes "representative deliberative processes." These processes include citizens' assemblies, citizen juries, and deliberative polling. They have in common selection through random or stratified random sampling (the contribution of this paper), but they differ in size and duration. Citizen juries, for example, are usually smaller (20-25 or so), and shorter in duration. Deliberative Polls are large (200-500), and typically of short duration—a weekend or two. Citizens' assemblies are

longer (several weekends up to as much as a year of weekends), and larger than citizen juries—in the range of 45 to 170 or so, usually as dictated by costs. The authors can simply substitute the term “deliberative minipublic” or “representative deliberative processes” for “citizens’ assembly” without any loss of meaning. If they need a reference, they might refer to the Smith and Setala chapter on deliberative minipublics in the Oxford Handbook of Deliberative Democracy.

Second, another bit of terminology the authors might use is “stratified random sampling,” which is simply a sample stratified by demographic categories. Again, no loss of meaning, as the authors are using the idea of a “quota category,” but “stratified random sampling” would be the term social scientists are most likely to use.

Third, as background, the authors might note that the justification for using stratified random sampling, in addition to biases introduced by the need for people to agree to be part of a randomly constituted pool from which deliberative minipublic members are selected, is that the categories should be relevant to the issue. So, for example, the British Columbia Citizens’ Assembly on Electoral Reform chose one man and one woman from each riding (electoral district) in British Columbia, as the topic dictated that ridings should be represented. Likewise, a post-hoc citizens’ assembly on Brexit stratified the sample by pro- and anti-Brexit, as it was important for both sides to be represented. While this aspect of deliberative minipublic composition has not been well theorized by most practitioners, it’s usually a consideration that makes it into the criteria for stratification. In some cases, an issue may require oversampling of a demographic group that is disproportionately affected.

Fourth, also as background, the authors might refer to Participedia.net, which is a semi-crowd-sourced website that documents democratic innovations. If they use the faceted search in “cases”, find the “recruitment method” options, and then do one search on “random selection” and another on “stratified random selection,” they will find a much more complete list than the OECD report upon which they rely (all the OECD cases are in Participedia). As of today, the search return 90 cases that are randomly selected, and 350 cases that use stratified random sampling.

Finally, the paper is a bit ad hoc about the “fairness” criterion for selecting an algorithm, which the authors assert is a driver of the (democratic) legitimacy of the process. Ideally, that would be the case. But we don’t actually know much about how broader publics view deliberative minipublics. From the bits of research we have (see, e.g., Cutler, et. al. in Warren and Pearse, *Designing Deliberative Democracy*), it appears that the representative qualities of minipublics are more important than the fairness of the selection process. Most people have no idea as to what random selection is, let alone have thoughts about its legitimacy. What people appear to care about is that the body looks like regular citizens, in contrast to bodies of comprised of professional politicians. We just need more research. I see there’s a new piece in *Representation* by James Pow: “Mini-Publics and the Wider Public: The Perceived Legitimacy of Randomly Selecting Citizen Representatives,” that concludes that people don’t focus on the particulars of selection processes. There are some survey experiments (e.g., the Pow piece). New survey research is coming (there is some EU money going in this direction), but we’re probably a few years away from results. So until we have more, the authors should be cautious about drawing a hard link between the fairness of selection processes and democratic legitimacy. I do agree, however, with their discussion of a cognate criterion, transparency. They note that some organizers use staged lottery-like mechanisms (which people do know about from lotteries!) to simulate fairness. They note, again importantly, that their algorithm can be staged in the same way.

Mark E. Warren
University of British Columbia

Referee #3 (Remarks to the Author):

This paper proposes an algorithm for sortition (random selection of representatives to form a panel) and compares its performance with that of a previous algorithm for the same problem (which the authors call LEGACY). The idea of sortition is to randomly select a fixed number of representatives from a pool of volunteers while maintaining demographic quotas. The proposed algorithm, LEXIMIN, outperforms LEGACY according to several fairness indices considered by the authors (including Nash social welfare and the Gini coefficient).

While sortition is a very interesting concept, it is a niche phenomenon. I would expect that many policy makers have reservations about an apportionment method which is solely based on proportionality and ignores competence. However, this is not the topic of the paper, and there is enough literature and research on sortition to justify the exploration of optimal algorithms for this problem. Also, I understand that sortition panels usually deliberate on topics rather than make decisions.

The paper reads well is and not too technical for non-specialists.

My main concern is that the contribution of the paper is a successful, but fairly standard, analysis of an algorithmic problem (including the NP-hardness proof) using standard optimization techniques (though the authors highlight the use of column generation for convex optimization).

The improvement over LEGACY is impressive, but to a large extent this seems to be due to the fact that this problem has never received a formal treatment by computer scientists or operations researchers who have the proper tools to address this problem. The authors write that they compare their approach to the LEGACY algorithm, because "it was the only one fully specified by an official implementation". According the authors, all previous approaches (including LEGACY) are iterative algorithms that select one representative after another (rather than trying to solve the entire problem to optimality). It should also be considered that, as far as I understand, LEGACY was not designed to maximize the minimal probability and similar fairness measures considered in the paper.

Some minor comments:

* Usually, "fairness notions" refer to concepts such as envy-freeness or proportionality (axioms that are either satisfied or not). I suggest to use the term "fairness index" instead.

* The example quota given on page 3 ("a 100-person panel contain between 49 and 50 women") seems rather narrow. Is this taken from a real-life example?

* The authors claim that there were only four previous deployed applications of fair division in the real world. How about the company Fair Outcomes, founded by Steven Brams, which says it licenses fair-division procedures?

* The fact that LEXIMIN satisfies equal treatments of equals and proportionality (Section 7 of the supplementary material) is well-known.

Reviewer 1

1.1 “I think it would be best to add details on the prior work on sortition by a subset of the authors.”

We added Section M12 (lines 828 – 881), in which we discuss the two prior papers on selection algorithms in detail.

1.2 “What is not entirely clear to me is whether fairness towards individual agents should be the main goal: (a) are agents eager to be selected or volunteer out of a sense of duty? (b) Is there evidence that maximizing some form of fairness, in the spirit formalized in the paper, is viewed as desirable in the broader sortition community?”

Before addressing the reviewer’s specific questions, we acknowledge and respond to their excellent broader point about the lack of clarity on why individual fairness should be the main goal. To address this in the revision more holistically, we reformulated the introduction to more prominently and explicitly provide evidence in favor of this objective. This reformulation includes a paragraph introducing the foundations of this objective in both the political theory literature and in more applied work on sortition (lines 58 – 67). We also further contextualized individual fairness among other sortition desiderata advocated in the literature in the new methods Section M14 (lines 918 – 1054).

Regarding (a): Participants opt into the pool for a wide range of reasons, often including both elements of personal eagerness and of a sense of duty. A nice glimpse into these motivations is given by a video by Of By For¹, in which pool members recorded their motivations right at the moment of joining the pool.

To the best of our knowledge, there are not many systematic investigations into participants’ motivations, and, in the examples we know of, the question of motivations is not clearly separated from participants’ attitudes towards the panel more broadly. Vincent Jacquet has asked participants of two citizens’ assemblies about their expectations going into the process [4], and finds evidence for both eagerness and a sense of duty as motivating factors. Specifically, the paper categorizes the expectations into two clusters: *internal expectations* (“gratifications relating to attendance of the mini-public”) and *external expectations* (“concerning the uptake of the mini-public in the broader political system”). In the area of internal expectations, two of the three subcategories of expectations (desire of sociability, learning) clearly point towards a personal eagerness to participate. While the third subcategory is “fulfilment of a civic duty,” pool members motivated by this sense of duty might still derive satisfaction of being chosen to serve.² To give support to the idea that people generally want to participate, we added references to this research and sources of participant testimony in line 50. In addition to these sources, Dr. Hennig’s extensive experience in recruiting citizens’ assemblies confirms that many people are eager to be selected and are disappointed if they are not.

The political theory literature also discusses the topic of eagerness versus duty (in slightly different terms) when considering the goal of equality. For example, Peter Stone makes the point that, whether holding political office is a *benefit* or a *burden*, it makes sense to share these benefits or

¹<https://vimeo.com/458304880#t=17m59s> from around 17:59 to 21:23.

²In an unrelated deliberative minipublic in the UK, participants expressed their sense of duty as wanting “to ‘make a contribution’, to ‘do something worthwhile’, to ‘make a difference’, to ‘benefit the future of our children’ and to ‘put something back’,” [3] which we understand to express an eagerness to be *personally* involved in an altruistic effort. Since the pool for this minipublic was not recruited from a random sample, we are cautious about generalizing the findings of Davies et al. [3] to minipublics with more rigorous recruiting methodologies, but we hope that these quotes illustrate the nuances in motivations summarized by the authors as “citizenship identity and public duty.”

burdens equally: “this argument for sortition requires no consensus on political office being a benefit or a burden. It merely requires that, whether office holding is a good or a bad, it is equally so for some group of people. Whenever this is the case, the just lottery rule applies” [8, p. 126].

Regarding (b): Our own experience with the quick uptake of our algorithm by practitioners demonstrates that many organizations see benefits in choosing an algorithm promoting individual fairness. We have also discussed our algorithms with a broader set of practitioners around the world, and, anecdotally, the algorithmic goal of individual fairness has been met with interest and enthusiasm.

In terms of written evidence, prominent groups of practitioners have put forward the ideal of equal selection probabilities as a desirable property of sortition in their guides instructing other organizations on how to carry out citizens’ assemblies. We added citations to two such guides in line 65: The first guide was published by MASS LBP in 2017, and states that “civic lotteries are built on the principle of sortition, or random selection, which maximizes fairness by ensuring that each person has an equal chance of being selected” [6]. The second guide was published by the Allianz Vielfältige Demokratie together with the Bertelsmann Foundation in 2018, and it states that an important benefit of sortition is that it “. . . guarantees equality of opportunity. With random selection, every citizen has the same probability of being selected” [1]. It is worth pointing out that both of these guides gloss over the fact that equal probabilities are rarely achieved in practice. Our work might help these organizations to align practice with their public messaging.

1.3 *“Another issue that I’d like to see discussed is intersectionality: is it possible to compare the algorithmic framework proposed in the paper to legacy algorithms in this regard? E.g., in one example considered in the paper the legacy algorithm is less likely than the fair algorithm to choose a conservative female, which is a rare combination of attributes. One would think that including many different combinations of attributes may be a relevant goal. Can the authors comment on the behavior of their algorithms with respect to this goal?”*

To address this point, we added methods Section M10 (lines 701 – 804), which contains an investigation of how accurately both algorithms represent intersectional groups.

A priori, we would expect neither LEGACY nor LEXIMIN to be well-positioned to proportionally represent these groups. LEXIMIN is designed to protect intersectional groups from extreme exclusion, but neither algorithm receives precise information on which share the intersectional groups have in the population. Nonetheless, it is an interesting question how a transition of LEGACY to LEXIMIN would impact the representation of intersectional groups. In Section M10, we investigate this question in one of our real-world datasets³, finding that both algorithms represent intersectional groups to essentially the same degree of precision (lines 736 – 747). We also do an additional preliminary analysis to offer a possible explanation for this similarity (lines 748 – 777). Finally, we discuss how our framework *can* be used to explicitly promote proportional representation of intersectional groups in the case where the population shares of these groups are known. In particular, we suggest two approaches for supporting the proportional representation of these groups within our framework: imposing hard constraints with quotas, or imposing soft constraints by incorporating the representation of intersectional groups into the optimization objective (lines 779 – 804).

³We make this comparison for only one dataset, $sf(e)$, because doing so requires knowing the rates of intersectional groups in the population underlying the panel, which requires finding a population-level survey with questions matching all quota features. We have such data for only this panel. Most of the other citizens’ assemblies analyzed in our paper happened on a regional or local level, which means that matching survey data are unlikely to exist.

Reviewer 2 (Prof. Warren)

2.1 *“First, the authors don’t quite have the covering terminology right. As the terms have developed, “citizens’ assemblies” are one kind of “deliberative minipublic,” which is the covering term.”*

In the revision, we have not yet fully adopted the term “deliberative minipublic” over “citizen’s assembly” because we feel that the term “citizen’s assembly”, while not referring to the whole breadth of participatory processes that our work applies to, is a more accessible term for a general audience. This perspective relies on Dr. Hennig’s experience organizing many sortition panels via the Sortition Foundation. To ensure that our meaning is clear to readers of all backgrounds, including those with expertise in sortition, we added footnote a to the paper’s first usage of “citizen’s assembly”. This footnote provides the names of other types of deliberative minipublics and clarifies that our results apply to the full breadth of deliberative minipublics. If Prof. Warren finds that these changes do not sufficiently address his concern, we will switch to using the term “deliberative minipublic” throughout the paper.

2.2 *“Another bit of terminology the authors might use is “stratified random sampling,” which is simply a sample stratified by demographic categories. Again, no loss of meaning, as the authors are using the idea of a “quota category,” but “stratified random sampling” would be the term social scientists are most likely to use.”*

Stratified sampling is indeed closely related to the quota constraints we consider, and Prof. Warren is very right that we should address this connection, especially given the prominence of stratified sampling in the political science literature. From a statistical point of view, however, stratified sampling can only be applied to quotas of a very restricted form, and, in fact, much of our algorithmic challenge arises from the fact that quotas as used in practice are not amenable to stratified sampling. Because the quotas we consider are more general, the terms “stratified sampling” and “quota categories” are not technically interchangeable. To provide more detailed information about our work’s connections to stratified sampling, we added a new methods Section M13 (lines 883 – 916). We also added footnote b to the body of the paper, which points readers to M13.

2.3 *“As background, the authors might note that the justification for using stratified random sampling, in addition to... is that the categories should be relevant to the issue.”*

We incorporated Prof. Warren’s point that quota categories often include features specifically relevant to the policy issue being discussed (line 57).

2.4 *“Also as background, the authors might refer to Participedia.net, which is a semi-crowd-sourced website that documents democratic innovations.”*

We cited Participedia.net with the appropriate filters applied (line 28).

2.5 *“The paper is a bit ad hoc about the “fairness” criterion for selecting an algorithm. ... Until we have more [research on how broader publics view deliberative minipublics], the authors should be cautious about drawing a hard link between the fairness of selection processes and democratic legitimacy.”*

We thank Prof. Warren for these excellent comments, which have pushed us to make more explicit the foundations of our approach in the political theory literature. We did so in the following ways: (1) First, we reformulated the introduction to more explicitly and prominently motivate the objective of individual fairness by connecting it with the ideal of *equality* widely promoted as an advantage of sortition (lines 58 – 67). The reformulated introduction also includes a discussion

of how our algorithm balances equality with the goal of *descriptive representation*: In particular, we preserve the quota-based approach to representativeness currently used, and only *subject to* achieving descriptive representation, do we maximize fairness (lines 52 – 57, 68 – 70). (2) Second, we added Section M14 (lines 918 – 1054), which more broadly discusses several desiderata of sortition from the political theory literature (equality, representativeness, efficiency, and protection against conflict and domination) that one could pursue in designing selection algorithms. For each goal, we discuss how it relates to the practical sortition setting and, if applicable, to our algorithm. In drawing a link between the fairness of the selection process and legitimacy, we were thinking along the lines of work by Courant [2] and Parker [7], both of which directly relate the equality of selection probabilities in sortition and (normative rather than descriptive) legitimacy. However, we fully acknowledge that we are not experts in this area. Thus, we removed both mentions of legitimacy contained within the original draft (one from the abstract, one from the discussion).

Reviewer 3

3.1 *“I would expect that many policy makers have reservations about an apportionment method which is solely based on proportionality and ignores competence. . . . Also, I understand that sortition panels usually deliberate on topics rather than make decisions.”*

Whereas elected officials certainly have reservations about the most radical proposals involving sortition, we would like to point out that less radical forms of sortition receive broader support among politicians. Recently, Jacquet, Niessen, and Reuchamps [5] have surveyed Belgian members of parliament about their support for different models of integrating deliberative minipublics into the political system. The model typically practiced right now, in which citizens’ assemblies are purely consultative, splits members of parliament relatively evenly between support (48%) and opposition (41%). Quite surprisingly, even the quite radical idea of a “mixed” legislative chamber (composed partially of elected representatives and of citizens selected by sortition) received a substantial amount of support among members of parliament (27%), and was favored by a majority of leftist politicians. Outside this specific study, important political figures have recently been pushing for (consultative) sortition panels, including French President Emmanuel Macron⁴, President of the European Commission Ursula von der Leyen⁵, former UK Prime Minister Gordon Brown⁶, and former UN Secretary General Kofi Annan⁷.

In his review, Prof. Warren gave an excellent explanation for what might drive politicians to support citizens’ assemblies: Despite their lack of competence requirements, citizens’ assemblies have an encouraging track record of reaching informed and broadly-supported conclusions [9], which distinguishes them from forms of democratic participation such as referenda. Moreover, citizens’ assemblies include a more representative set of citizens than purely self-selected forms of participation. Thus, when politicians have a reason to hand some amount of control to the population (for example, as a response to the yellow vests movement or out of a desire to add democratic elements to the European Union without changing the treaties), citizens’ assemblies might be an attractive option.

3.2 *“My main concern is that the contribution of the paper is a successful, but fairly standard,*

⁴<https://www.newdemocracy.com.au/2017/06/20/french-presidential-election-and-sortition/>

⁵<https://carnegieeurope.eu/2019/11/26/new-wave-of-deliberative-democracy-pub-80422>

⁶<https://www.theguardian.com/commentisfree/2019/jan/20/citizens-assembly-brexit-article-50-britain>

⁷<https://www.kofiannanfoundation.org/supporting-democracy-and-elections-with-integrity/athens-democracy-forum/>

analysis of an algorithmic problem (including the NP-hardness proof) using standard optimization techniques (though the authors highlight the use of column generation for convex optimization)."

In our view, our main contribution to computer science does not lie in our algorithmic analysis, and we clarify one of our more important contributions to computer science in the next paragraph. The reviewer's comments made us realize, however, that the original version of the paper did not sufficiently highlight the generality and broader applicability of our algorithmic results. To address this point, we extended Section M3 (lines 414 – 476) to explicitly discuss how our column-generation approach can be seen as solving a general class of convex programs. Moreover, we discuss how our algorithms immediately apply to settings outside of sortition, including kidney exchange (lines 460 – 476).

While we do, as reviewer 3 points out, make a novel algorithmic contribution, we do not see this as our main contribution. Rather, a more important contribution of this work is its translation of a central, practical, and seemingly abstract problem in sortition into a mathematically precise form that enables the application of optimization techniques. This work prompts new questions in multiple fields, and in particular opens up several new technical problems for computer scientists: For example, in the discussion, we raise the question of how to discretize panel distributions, motivated by potentially greater transparency (lines 250 – 252); and in the new Section M14, we pose the question of how to sample panels with maximum entropy, motivated by the yet-unresolved issue of discouraging manipulation in practical sortition (lines 1039 – 1043). In a broader sense, we hope that our work will encourage algorithms researchers to engage with political scientists and practitioners on questions of civic participation, facilitating the identification of new problems that offer opportunities for technical innovations with high real-world impact.

3.3 *"The authors write that they compare their approach to the LEGACY algorithm, because "it was the only one fully specified by an official implementation": According the authors, all previous approaches (including LEGACY) are iterative algorithms that select one representative after another (rather than trying to solve the entire problem to optimality). It should also be considered that, as far as I understand, LEGACY was not designed to maximize the minimal probability and similar fairness measures considered in the paper."*

The reviewer is correct that LEGACY is not designed to maximize the minimal probability, or other fairness measures considered in this paper. In fact, with regards to the objective of maximizing the minimal probability, we know our algorithm is *optimal*, so the question is not whether we beat the existing algorithm — it is how much we improve upon what was previously done in practice. We reformulated our description of the benchmark (lines 173 – 176) to more prominently emphasize the objective of this comparison. This reformulation also more clearly states our reasons for selecting LEGACY as our benchmark over other existing algorithms. We also point out that we do demonstrate that our algorithms are more holistically fair than existing algorithms (rather than just better on the precise notion LEXIMIN optimizes and LEGACY does not), by comparing LEXIMIN and LEGACY on the basis of other fairness measures (the Gini coefficient and the geometric mean). There are trade-offs between different fairness measures, so, unlike on the notion of minimum probability, it is not clear a priori that, on these other measures, LEXIMIN would do better. Nonetheless, we find that LEXIMIN also performs substantially better across all instances (and substantially better on most instances) according to these other fairness measures (lines 224 – 233).

3.4 *"Usually, 'fairness notions' refer to concepts such as envy-freeness or proportionality (axioms that are either satisfied or not) I suggest to use the term 'fairness index' instead."*

We changed all uses of the term “fairness notion” to the term “fairness measure” (which in some places fits more seamlessly into the text than “fairness index”).

3.5 “The example quota given on page 3 (‘a 100-person panel contain between 49 and 50 women’) seems rather narrow. Is this taken from a real-life example?”

While not taken from any specific panel, the level of tightness in the example quota we originally gave (“quotas might require a 100-person panel to contain between 49 and 50 women”) is in fact quite common in practice. We did change the example to be at a more typical scale, since most panels are smaller than 100 people: “quotas might require that a 40-person panel contain between 20 and 21 women” (lines 55 – 56).

3.6 “The authors claim that there were only four previous deployed applications of fair division in the real world. How about the company Fair Outcomes, founded by Steven Brams, which says it licenses fair-division procedures?”

The original version of the paper referenced the adjusted winner website, which is operated by Fair Outcomes (<https://www.fairproposals.com/about-us>) and is their only application that is freely accessible to the public. We updated the reference in our list of deployments of fair division to refer to Fair Outcomes more broadly.

3.7 “The fact that LEXIMIN satisfies equal treatments of equals and proportionality (Section 7 of the supplementary material) is well-known.

We, of course, agree that these results are not surprising, and we did not intend to imply that proving them would constitute a significant contribution. That being said, properties of the form “[algorithm] satisfies [axiom]” are always proved in the context of a specific model, and might hold in one model but not the other. For this reason, we did not want to leave the statements without proof, especially given that we do not expect all readers to be experts in fair division. Note also that, while the proof of proportionality follows the standard outline one would expect, this is due to the fact that we have chosen the appropriate way of *defining* proportionality in our context, out of multiple possible definitions. For added clarity, we added an explicit note that these proofs follow standard arguments in the fair division literature (lines 1133 – 1134, M17).

References

- [1] Allianz Vielfältige Demokratie and Bertelsmann Foundation. Citizens’ participation using sortition, 2018. http://aei.pitt.edu/102678/1/181102_Citizens__Participation_Using_Sortition_mb_-_Copy.pdf.
- [2] D. Courant. Sortition and democratic principles: A comparative analysis. In *Legislature by Lot: Transformative Designs for Deliberative Governance*, The Real Utopias Project. Verso, 2019.
- [3] C. Davies, M. S. Wetherell, and E. Barnett. *Citizens at the Centre: Deliberative Participation in Healthcare Decisions*. Policy Press, Clifton, Bristol, UK, 2006.
- [4] V. Jacquet. The role and the future of deliberative mini-publics: a citizen perspective. *Political Studies*, 67(3):639–657, 2019.
- [5] V. Jacquet, C. Niessen, and M. Reuchamps. Sortition, its advocates and its critics: An empirical analysis of citizens’ and MPs’ support for random selection as a democratic reform proposal. *International Political Science Review*, page 0192512120949958, 2020.

- [6] MASS LBP. How to run a civic lottery: Designing fair selection mechanisms for deliberative public processes, 2017. <https://static1.squarespace.com/static/6005ceb747a6a51d636af58d/t/6010cf8f038cf00c5a546bd7/1611714451073/civiclotteryguide.pdf>.
- [7] J. M. Parker. *Randomness and Legitimacy in Selecting Democratic Representatives*. PhD thesis, University of Texas at Austin, 2011.
- [8] P. Stone. *The luck of the draw: The role of lotteries in decision making*. Oxford University Press, 2011.
- [9] D. Van Reybrouck. *Against Elections: The Case for Democracy*. Random House, 2016.

Reviewer Reports on the First Revision:

Referee #1:

Remarks to the Author:

I am satisfied with the revision.

Referee #2:

Remarks to the Author:

The authors have made substantial improvements to the original manuscript, and I'm pleased to support publication with a minor revisions. The caveat still applies that I am a political theorist, and so lack the skills to understand critically the technical parts of the argument—though the natural language explanations are clear and make sense.

1. "Citizens' Assemblies": I would still prefer the more technically accurate term "deliberative minipublics," especially since this is a covering term that applies to all of the kinds of processes covered by this selection algorithm—those that are demographically representative (hence, mini-public) and deliberative. There is some worry that, for academics, the "citizens assembly" term will feel overly narrow, for while non-academics, the emerging technical meaning of "citizens assembly" will be lost to its common sense meaning, namely, any assembly of citizens. I expect that there is some division between practitioners and academics, with the academics having settled on "deliberative minipublics," and practitioners using these terms more loosely. It would be nice to be more definitive about the terminology, but this is a rapidly developing field with terminology that is rapidly evolving. With these considerations in mind, the authors should use their judgment.

2. The problem created by unequal selection into pools from which participation isn't stated as straightforwardly as it should be in the abstract or the beginning of the paper. Thus, the abstract states that "citizens would ideally be selected to serve on this panel with equal probability" but that "this is impossible due to demographic quotas." This formulation makes the quotas sound like a defect that needs to be overcome, when, in fact, this is the point of a representative panel. It's a feature, not a bug. Along the same lines, it would be good if the authors could explain, both in the abstract and introduction, why equal selection from a pool of volunteers wouldn't produce a demographically representative panel. It's clear to all of us who follow these questions, but for a more general audience, particularly outside of the social sciences, it might not be obvious that those who volunteer into a pool will almost always be older, more educated, more likely to be fluent in the dominant language, and more likely to belong to a dominant race or ethnicity. The point of the algorithm is to generate more equal probabilities of being selected with the quota constraints aiming for representativeness. These two goals are clear at the end of the paper (M14ff), but should be clear at the beginning, as these are constitutive of deliberative minipublics.

Referee #3:

Remarks to the Author:

Thanks for your response to my questions. I appreciate the changes that have been made to the paper.

I agree that the main contribution is not algorithmic, but a mathematical formalization of the sortition problem which enables the application of optimization techniques. I maintain my point that the success of the proposed algorithm is largely due to the fact the problem has never received a formal treatment by computer scientists or operations researchers. Of course, this is not your fault and has to do with the fact that sortition has not yet seen widespread acceptance. Your results can certainly help to make sortition more popular.

Author Rebuttals to First Revision:

Reviewer 2

Point #1: “Citizens’ Assemblies”: I would still prefer the more technically accurate term “deliberative minipublics,” especially since this is a covering term that applies to all of the kinds of processes covered by this selection algorithm—those that are demographically representative (hence, mini-public) and deliberative. There is some worry that, for academics, the “citizens assembly” term will feel overly narrow, for while non-academics, the emerging technical meaning of “citizens assembly” will be lost to its common sense meaning, namely, any assembly of citizens. I expect that there is some division between practitioners and academics, with the academics having settled on “deliberative minipublics,” and practitioners using these terms more loosely. It would be nice to be more definitive about the terminology, but this is a rapidly developing field with terminology that is rapidly evolving. With these considerations in mind, the authors should use their judgment.

Response: We elect to use the phrase “citizens’ assembly” on the recommendation of practitioners, but we modify the first paragraph of the introduction to prominently state that deliberative minipublics encompass citizens’ assemblies and several similar forms of deliberative democracy (e.g., citizens’ juries). Our hope is that this more definitive and prominent statement of this term will make our exposition and the breadth of our work’s implications clear to academics.

Point #2: The problem created by unequal selection into pools from which participation isn’t stated as straightforwardly as it should be in the abstract or the beginning of the paper. Thus, the abstract states that “citizens would ideally be selected to serve on this panel with equal probability” but that “this is impossible due to demographic quotas.” This formulation makes the quotas sound like a defect that needs to be overcome, when, in fact, this is the point of a representative panel. It’s a feature, not a bug. Along the same lines, it would be good if the authors could explain, both in the abstract and introduction, why equal selection from a pool of volunteers wouldn’t produce a demographically representative panel. It’s clear to all of us who follow these questions, but for a more general audience, particularly outside of the social sciences, it might not be obvious that those who volunteer into a pool will almost always be older, more educated, more likely to be fluent in the dominant language, and more likely to belong to a dominant race or ethnicity. The point of the algorithm is to generate more equal probabilities of being selected with the quota constraints aiming for representativeness. These two goals are clear at the end of the paper (M14ff), but should be clear at the beginning, as these are constitutive of deliberative minipublics.

Response: We thank Professor Warren for this excellent comment. We have modified the abstract and the introduction to explicitly place the two goals, quotas (representation) and equal probabilities, on even footing in terms of importance. We also point out in the abstract and in more detail in the introduction that the pool tends to be unrepresentative of the population, tending for example to overrepresent those who are more highly educated. Finally, we explicitly explain in the intro that the under- and overrepresentation of certain groups in the pool puts our two goals, of representation and equality, in tension, and ultimately precludes selecting people with equal probabilities while also producing a representative panel.